# Direct Identification of Urinary Tract Pathogens by MALDI-TOF/TOF Analysis and De Novo Peptide Sequencing

**DOI:** 10.3390/molecules27175461

**Published:** 2022-08-25

**Authors:** Ema Svetličić, Lucija Dončević, Luka Ozdanovac, Andrea Janeš, Tomislav Tustonić, Andrija Štajduhar, Antun Lovro Brkić, Marina Čeprnja, Mario Cindrić

**Affiliations:** 1Novo Nordisk Foundation Center for Biosustainability, Technical University of Denmark, 2800 Lyngby, Denmark; 2Division of Molecular Medicine, Ruđer Bošković Institute, Bijenička 54, 10000 Zagreb, Croatia; 3Clinical Department of Laboratory Diagnostics, University Hospital Dubrava, Avenija Gojka Šuška 6, 10000 Zagreb, Croatia; 4Conscius Ltd., Kutnjački put 9, 10000 Zagreb, Croatia; 5Division for Medical Statistics, Andrija Štampar Teaching Institute of Public Health, Mirogojska cesta 16, 10000 Zagreb, Croatia; 6Institute of Physics, Bijenička cesta 46, 10000 Zagreb, Croatia; 7Special Hospital Agram, Agram EEIG, Trnjanska cesta 108, 10000 Zagreb, Croatia

**Keywords:** uropathogenic infection, tandem mass spectrometry, de novo peptide sequencing, peptide identification software

## Abstract

For mass spectrometry-based diagnostics of microorganisms, matrix-assisted laser desorption ionization time-of-flight mass spectrometry (MALDI-TOF MS) is currently routinely used to identify urinary tract pathogens. However, it requires a lengthy culture step for accurate pathogen identification, and is limited by a relatively small number of available species in peptide spectral libraries (≤3329). Here, we propose a method for pathogen identification that overcomes the above limitations, and utilizes the MALDI-TOF/TOF MS instrument. Tandem mass spectra of the analyzed peptides were obtained by chemically activated fragmentation, which allowed mass spectrometry analysis in negative and positive ion modes. Peptide sequences were elucidated de novo, and aligned with the non-redundant National Center for Biotechnology Information Reference Sequence Database (NCBI*nr*). For data analysis, we developed a custom program package that predicted peptide sequences from the negative and positive MS/MS spectra. The main advantage of this method over a conventional MALDI-TOF MS peptide analysis is identification in less than 24 h without a cultivation step. Compared to the limited identification with peptide spectra libraries, the NCBI database derived from genome sequencing currently contains 20,917 bacterial species, and is constantly expanding. This paper presents an accurate method that is used to identify pathogens grown on agar plates, and those isolated directly from urine samples, with high accuracy.

## 1. Introduction

In clinical microbiology, methods and techniques for reliable, direct identification of pathogens in human body fluids are continuously investigated. Mass spectrometry-based methods, especially matrix-assisted laser desorption ionization time-of-flight mass spectrometry (MALDI-TOF MS), have been successfully used in routine clinical characterization of pathogens for the past decades [1]. In most cases, competing methods for microbial identification include 16S ribosomal rRNA gene sequencing [2] and real-time polymerase chain reaction (RT-PCR) [3]. Although 16S rRNA sequencing is a widely used method to study the phylogenetic relationships among bacteria, it has low discriminatory power between related species [4]. Another DNA-based method commonly used in clinical practice to identify pathogens is the RT-PCR method. It is a rapid and sensitive method that allows high-throughput analysis, and even the quantification of pathogens [5]. However, the primers of RT-PCR target one or more bacterial species, so non-targeted pathogens cannot be identified. Spectroscopic methods such as Fourier transform near-infrared (FT-NIR) [6] and Raman spectroscopy [7] are also applicable for pathogen identification. These methods are rapid, non-invasive, inexpensive, and, therefore, suitable for routine quality control analyzes. However, the aforementioned technologies are limited by low sensitivity, a relatively small number of reference species (approximately 100), strong background interferences (e.g., water), and, finally, the cultivation of microorganisms [8]. Moreover, the interpretation of the spectral data obtained is often ambiguous, and spectroscopic instruments must be coupled with other standard methods to ensure their reliability. Despite all the advantages and disadvantages of the alternative methods, MALDI-TOF MS is the predominant and most reliable technique in clinical practice for the rapid and cost-effective identification of a wide range of pathogens [9].

Standard urine culture analysis is the most sought-after test in clinical laboratories, as urinary tract infections (UTI) affect more than 150 million people worldwide each year. The most common pathogens of UTIs are uropathogenic *Escherichia coli*, followed by *Klebsiella pneumoniae*, *Proteus mirabilis*, *Staphylococcus saprophyticus*, *Enterococcus faecalis*, Group B *Streptococcus*, *Pseudomonas aeruginosa*, *Staphylococcus aureus*, and *Candida species* [9]. The main disadvantage of urine-culture-based identification methods is the long cultivation time (36–72 h) and possible non-specific identification [10]. Due to the high prevalence of urinary tract infections mentioned earlier, improvement of identification methods in terms of speed, accuracy, and more meaningful identification is an ongoing need in clinical microbiology research. Several bottom-up proteomic studies concerning MS/MS microbial identification were published. In these studies, liquid chromatography–tandem mass spectrometry (LC–MS/MS) and database search tools were used to identify pure microbial cultures [11,12,13,14,15]. Using LC–MS/MS and various database search tools, researchers compared experimentally obtained MS/MS spectra of analyzed peptides with the in-silico-generated peptide fragmentation spectra in the protein database. Although the LC–MS/MS approach to microorganism identification results in accurate and robust species identification, the protein databases used often encounter obstacles due to the small number of available species. The standard identification procedure usually consists of the step of bacterial cultivation, MALDI-TOF MS analysis of proteins in the mass range of 2–20 kDa, and spectral alignment of the obtained signals with the reference spectral library. To speed up the process, and avoid a time-consuming cultivation step, several studies adapted a MALDI-TOF MS method to identify pathogens directly from common clinical samples such as urine [10,16,17,18,19,20] or a positive blood culture [21,22,23]. However, direct analysis of samples may be subject to low confidence levels due to polymicrobial or human protein interference [17]. In addition, a recent article described a method for direct identification of UTIs using tandem mass spectrometry, machine learning [24], and bottom-up metaproteomic analysis [25].

A widely used, high-resolution LC–MS/MS technique for proteomic analysis is not a standard in clinical research because of the need for highly skilled personnel, long analysis time, and high maintenance costs [26]. Our method utilizes a MALDI-TOF/TOF MS instrument that overcomes some of the limitations of MALDI-TOF MS and LC–MS/MS strategies. Compared with LC–MS/MS, MALDI-TOF MS and MALDI-TOF MS/MS instruments are much easier to integrate into hospital pipelines, because of their relative simplicity of use and lower operating costs. In addition, the MALDI MS technique is less susceptible to contamination than LC–MS/MS [27]. Unlike MS, MS/MS analysis allows sequencing of each peptide individually, so major interferences from human proteins or other protein contaminants do not interfere with bacterial identification. From this point of view, the optimal tool for pathogen identification in clinical microbiology should include the analytical methods MALDI MS and MS/MS. Therefore, the newly developed de novo sequencing identification method in this research is a complementary improvement to the currently available identification techniques.

We present here a novel bottom-up proteomics approach for the direct identification of microorganisms from a urine sample by tandem mass spectrometry using the MALDI-TOF/TOF MS instrument and de novo peptide sequencing (Figure 1). The method is based on in-solution *N*-terminal tryptic peptide derivatization by 4- or 5-formyl-1,3-benzenedisulfonic acid (FBDSA), which is described in detail elsewhere [28,29,30]. Derivatization allows de novo sequencing of positive and negative peptide mass spectra obtained after analysis of MALDI-MS/MS. Newly developed software called Protein Acrobat calculates the most probable peptide sequences from the aligned negative MS/MS and positive MS/MS mass spectra using a graph-based algorithm. The resulting list of probable peptides was aligned with the NCBI*nr* database using the protein basic local alignment search tool (BLASTp) [31], and the results were scored according to the expected (E) value. Derived peptides above the predefined E-value threshold were assigned to a species in the database using a peptide-matching algorithm based on the Apache Lucene search engine [32]. The presented positive and negative de novo peptide sequencing approach was applied to the identification of UTI pathogens directly from urine samples.

## 2. Results

### 2.1. Results of Standard Urine Culture Test and Identification of Urine Bacterial Isolates Grown on Blood Agar Plates

A standard urine culture test, described in detail in Materials and Methods, was used to determine the number of viable bacteria in urine specimens from patients with a confirmed UTI. The biochemical test “European Guidelines for Urinalysis” [33] was applied to determine the causative agent of the infection and later comparison with the results of mass spectrometry performed in this research. Urine samples from 33 patients were collected on a workday (medium-sized hospital) using the clean-catch method and yielded a bacterial count of 10^5^ colony-forming units (CFU/mL) or more. The samples were used for comparison between direct MALDI-TOF/TOF MS UTI identification and standard urine culture test.

The microorganisms from 33 collected urine samples were previously identified by standard urine culture test as: *E. coli* (16 samples), *P. aeruginosa* (4 samples), *E. faecalis* (3 samples), *K. pneumoniae* (3 samples), *P. mirabilis* (3 samples), and 1 sample each of *Citrobacter freundii*, *Stenothropomonas maltophilia*, *Candida albicans*, and *Klebsiella aerogenes*.

Proof of concept for the MALDI-TOF/TOF MS method for direct identification of UTIs was confirmed and validated on pure urine colonies of *E. coli*, *P. mirabilis*, *P. aeruginosa*, and *K. pneumoniae* grown on blood agar plates. For each designated species, ten biological replicates were processed and analyzed using MALDI-TOF/TOF MS as described in Materials and Methods. Tandem mass spectra were acquired in positive and negative ion modes, and peptide sequences were determined using the Protein Acrobat de novo sequencing algorithm.

### 2.2. Protein Acrobat De Novo Sequencing Algorithm

The data obtained after positive and negative MS/MS analysis in mascot generic format (MGF) were processed using Protein Acrobat. First, the b-ion series were converted from MS/MS negative to y-ions in MS/MS positive, following a peptide fragmentation rule that the sum of the nominal *m/z* of the b-ions and the corresponding y-ions equals the nominal value of the precursor mass and the mass of two hydrogen atoms:*m/z* (y) = MH^+^ − *m/z* (b) + H^+^(1)

Both spectra were normalized to account the lower peak intensities in the positive spectrum, which was achieved by dividing the spectra by their respective peak maxima. Peaks with a mass within a specified range (0.01 Da) were merged into a single spectrum to remove redundant peaks from positive and negative MS/MS spectra. Figure 2 shows an example of processing positive and negative MS/MS spectra of the peptide belonging to the putative stress response protein (accession number STS794191) found in *K. pneumoniae* species. The resulting mass spectrum was converted into a graphical spectrum, in which the nodes represent *m/z* values. The weighting of each node corresponds to the normalized intensity value. Edges were formed between nodes with a mass difference of any amino acid mass. Dynamic programming techniques were used to find paths between the starting node (zero mass) and the ending node (MS ion *m/z* mass), using the sum of the maximum node weights as the path score. All possible amino acid sequences were determined and scored relative to the overall peak intensity of fragmented ions. The ten highest scoring peptide sequences were matched to a filtered NCBI nr database using the BLASTp tool integrated with Protein Acrobat. The filtered NCBI nr database consisted of 48 common uropathogenic species adapted from Pinault et al. 2019 [11]. The complete list of the 48 bacteria is shown in Appendix A. The E-value was set in the range of 0.001–1, and was 0.1 depending on how many peptides matched the corresponding species.

Figure 3 shows graphical results of peptide matches for pure colony samples of *E. coli*, *P. mirabilis*, *K. pneumoniae*, and *P. aeruginosa*. The average number of peptides matched to the corresponding bacterial species is 129 for *E*. *coli*, 42 for *P. mirabilis,* 54 for *K. pneumoniae,* and 29 for *P. aeruginosa*. The data shown in the graph are combined from 10 analyses per species, and normalized to the sum of identified peptides. It shows the probability of the identified species as a function of the number of matched peptides. To simplify the plot, only the first five species with the most peptide matches are shown in the graph.

### 2.3. Direct Identification of Uropathogenic Bacteria from Urine Samples

To determine the accuracy of the UTI identification results, decision tree learning was used to define the confidence level of bacterial identification. For simplicity, the input dataset was obtained from the analysis of pure colonies of *E. coli*, *P. mirabilis*, *K. pneumoniae*, and *P. aeruginosa*. The original dataset of 40 measurements (10 replicates for each bacterial species) was expanded using the mixup technique [34]. An expanded dataset defines the confidence level of direct bacterial identification of urine samples as follows: high confidence, medium confidence, and low confidence, denoted by the numbers 3, 2, and 1, respectively. Of the 33 urine samples analyzed, 29 samples are successfully identified. Protein Acrobat results for all urine samples are shown in Table 1. A high-confidence level (score 3) is reported for 22 samples, a medium-confidence level (score 2) for 7 samples, and a low = confidence level (score 1) for 4 samples. The low-confidence level samples are misidentified in three of four cases; *K. pneumoniae* and *C. freundii* are identified as *E. coli* and *K. pneuomoniae*, respectively. The remaining low-confidence level sample identified as *K. aerogenes* by the standard urine culture test is not identified by the direct identification method. In other words, it has an equal number of peptide matches with *K. aerogenes*, *K. pneumoniae*, *Enterobacter kobei*, and *Enterobacter cloacae* species. On the other hand, all samples with a medium-confidence level are correctly matched. Only 1 high-confidence level sample is misidentified (*Citrobacter freundii*), and the remaining 21 high-confidence level samples match the results of the standard urine culture test. The direct bacterial identification pipeline utilizing MALDI-MS/MS and Protein Acrobat is graphically depicted in Appendix A.

## 3. Discussion

Pathogen identification using tandem mass spectrometry is a developing methodology with promising results. Accurate identification of samples derived from pure colonies is shown in several publications [13,35]. However, the main advantage of this methodology is the possibility to directly use clinical samples, thus, avoiding the lengthy culture step [36]. Direct identification analysis was performed on urine samples with a bacterial count of 10^5^ CFU/mL or greater. Accurate identification at bacterial counts less than 10^5^ CFU/mL can be achieved using more sensitive nano LC-ESI–MS/MS instruments [24]. However, it is shown that accuracy in the aforementioned studies increases dramatically when a larger amount (>10^5^ CFU/mL) of bacteria is present in the urine. In fact, high resolution nanoLC-ESI–MS/MS allows the identification of a larger number of peptides, especially low abundance peptides, resulting in better species discrimination and, thus, better identification.

Extracted proteins from bacterial cultures and urine samples were digested prior to identification. The resulting complex peptide mixtures were separated into 36 fractions prior to mass spectra analysis, to increase proteome coverage and facilitate identification of low abundance proteins [37]. As MALDI-TOF/TOF MS is not coupled to a liquid chromatography system, peptides were separated offline in two dimensions using micro-chromatography cartridges. A strong anion exchanger, specifically a quaternary ammonium cation (QMA), was chosen as the stationary phase in the first dimension because peptide derivatization with 4-formyl-1,3-benzenedisulfonic acid (Materials and Methods, Section 4.4.), as described, enhances the binding of the peptides to the functional groups of the QMA stationary phase. Six eluted fractions in the first dimension were further separated using a SCX stationary phase, to achieve greater separation of peaks in MS analysis. Using a combination of anion and cation exchange, the samples were fractioned and desalted at the same time, producing a clean sample read for the MS analysis. Although direct UTI identification was achieved in ≤24 h, the protocol can be further accelerated by rapid trypsin digestion [38,39] and MALDI-TOF/TOF MS instruments with 2000 Hz laser repetition rates (10× faster acquisition).

Pure bacterial cultures of *E. coli*, *K. pneumoniae*, *P. mirabilis*, and *P. aeruginosa* were analyzed in ten replicates to assess the qualitative differences of the identified peptides. Although the method was validated using pure cultures, this study aimed to develop a rapid and accurate protocol for the identification of pathogens directly from urine samples, without prior cultivation. Since direct analysis results in an overall lower peptide count than pure cultures, a classification model was applied to evaluate the confidence level of species identification. It is found that 87.8% of the samples could be correctly assigned with high- and medium-confidence levels. Misidentifications occur in samples with low-confidence levels and in one sample with a high-confidence level. The incorrect results involve three samples of *Klebsiella* species and one *C. freundii*. The results could be explained by the mixed-species samples (e.g., two or more species in the sample), contaminated urine samples (e.g., large number of human cells), or the high genomic similarity between *Klebsiella*, *Escherichia*, and *Citrobacter* species [40,41], which makes them difficult to distinguish when only a small number of peptides are identified.

However, it is worth mentioning that the identification of *Klebsiella* species, which was challenging in direct analysis, was unambiguous when analyzing pure colonies. The identification method was based on de novo sequencing of tryptic peptides, and subsequent peptide sequence alignment with 48 common uropathogenic bacterial species present in the NCBI*nr* database (Appendix A). De novo peptide sequencing is a promising identification tool in proteomics that has been investigated over the past 20 years [42]. Despite various de novo sequencing algorithms and extensive research, there is an apparent lack of wider application, especially in the identification of complex protein mixtures [43,44]. This can be attributed to a few drawbacks of MS/MS spectra sequencing such as insufficient mass accuracy relative to non-linear sequences data structure consisting of determined amino acid at nodes and edges (e.g., an average MS/MS spectrum could be branched by the algorithm in several hundred sequences), and uneven fragmentation patterns caused by missing fragment ions. The named drawbacks, along with the noise present in the MS/MS spectra (Appendix A), hamper the complete detection of long sequences (larger than 20 amino acids), and contribute to sequence ambiguities. Disulfonic peptide derivatization at the *N*-terminus improves amino acid sequence recognition, by allowing the revelation of only b- ions in negative ion mode MS/MS, and y- ions in positive ion mode MS/MS [45]. The de-novo-identified peptides were aligned with protein sequences in the database, and microorganism species were determined based on the highest number of matched peptides. The described identification method can also be used to identify other pathogens in clinical, environmental, or food samples. However, rare pathogens or other species of interest could be identified by searching the entire NCBI*nr* database, which increases the search time but, on the other hand, allows the identification of virtually any species in the database.

## 4. Materials and Methods

### 4.1. Raw Materials

Sodium dodecyl sulphate (SDS; ≥99.9%), sodium dihydrogen phosphate (≥99.9%), formic acid (>98%), hydrochloric acid (>36.5%), DL-dithiothreitol (DTT; ≥99.9%), urea (≥99.9%), cyano-4-hydroxy cinnamic acid (CHCA; ≥99.9%), proteomic grade of trypsin from porcine pancreas, and 4-formyl-1,3-benzenedisulfonic acid disodium salt hydrate (97%) were obtained from Sigma-Aldrich (St. Louis, MO, USA); sodium cyanoborohydride from G-Biosciences (St. Louis, MO, USA); and ammonium bicarbonate (≥99.9%), acetonitrile (≥99.9%), potassium dihydrogen phosphate (≥99.9%), Tris base (≥99.9%), and Microcon-30 kDa 0.5 mL centrifugal filter devices were purchased from Merck Millipore (Burlington, MA, USA). Sep-Pak Accell Plus Vac cartridges as QMA stationary phase was obtained from Waters (Milford, MA, USA), and SCX stationary phase in bulk and resin-free AssayMAP cartridges were obtained from Agilent Technologies (St. Clara, CA, USA).

### 4.2. Urine Sample Collection Ethics

Clean-voided, midstream urine specimens were collected from outpatients who were referred for urine culture and analysis to the Department for Clinical Microbiology and Hospital Infections, University Hospital Dubrava, Zagreb, Croatia. The urine samples were tested for bacterial growth on culture plates. The number of bacteria was determined by colony counting on an agarose MacConkey plate, and the species were determined by biochemical identification, according to standard procedures [33].

### 4.3. Protein Extraction

Bacterial colonies from agar plates were collected with a microbiological loop and mixed with 400 µL of 50 mM ammonium bicarbonate. The bacterial suspension was centrifuged at 8000× *g* for 10 min at 4 °C to wash the cells, and the supernatant was discarded. Proteins extracted directly from urine samples were obtained by the following protocol: a volume of 15 mL of thawed urine sample was centrifuged at 500× *g* for 1 min at room temperature, to separate microbial cells from larger particles such as salt crystals and leukocytes. The supernatant was centrifuged at 4500× *g* for 30 min at 4 °C to obtain bacterial pellet, which was then resuspended in 400 µL of 50 mm ammonium bicarbonate, and again centrifuged at 8000× *g* for 10 min at 4 °C. The wash step with ammonium bicarbonate was repeated one more time. From this point on, further sample preparation protocol and analysis were the same for bacterial biomass obtained from pure colonies and urine samples.

Cell lysis was achieved by resuspending the bacterial pellet in 200 µL of lysis buffer containing 2% of sodium SDS, 100 mm Tris/HCl pH 7.8, and 0.1 M DTT. After incubation at room temperature for 10 min, cells were disrupted by sonication using the Sonoplus mini20 ultrasonic homogenizer (Bandelin, Berlin, Germany). Samples were sonicated for five cycles of 1 min, with a 30 s cooling interval between cycles, and the sonication amplitude was set to 50%. Afterwards, samples were placed into boiling water for 2 min to inhibit protease activity. The cell lysate was removed from each sample by centrifugation at 16,000× *g* for 20 min at 4 °C, and the protein fraction contained in supernatant was further processed.

### 4.4. Protein Purification and Digestion

The FASP protocol was adapted from Yu et al., and conducted with slight modifications [46]. In brief, the Microcon centrifugal filter devices were rinsed with 200 μL of ultrapure water and 200 μL urea solution (8 M urea in 100 mm Tris/HCL buffer). An aliquot of 100 µL of extracted protein mixture was added to a filter device, mixed with 200 μL of urea solution, and centrifuged at 12,000× *g* for 20 min at 20 °C. Flow through was discarded, and 200 μL of urea solution was added to a filter and spun at 12,000× *g* for 15 min at 20 °C. This step was repeated, and filter was further washed twice with 200 μL of ammonium bicarbonate buffer. Tryptic digestion was performed overnight by adding 2 μL of 1 mg/mL trypsin solution and 100 μL of ammonium bicarbonate buffer to the filter unit, and incubating the resulting solution overnight on a Comfort thermomixer (Eppendorf, Hamburg, Germany) at 37 °C and 500 rpm. After digestion, peptides were centrifuged of the filter device at 12,000× *g* for 15 min, followed by one wash step with 50 μL of ammonium bicarbonate buffer. The peptide solution was dried in vacuum concentrator at 30 °C.

### 4.5. Derivatization Procedure

Peptide derivatization procedure was previously described [29], and repeated here with modifications. Each sample containing dried tryptic peptides was reconstituted with 30 μL of derivatization solution. The derivatization solution contained 12.5 mM of 4-formyl-1,3-benzenedisulfonic acid disodium salt hydrate and 95.5 mm of sodium cyanoborohydride dissolved in 10 mm potassium dihydrogen phosphate pH 5. The derivatization was performed in a household microwave oven at 180 W for 8 min.

### 4.6. Two Dimensional Fractionation of Peptide Mixture

Sample fractionation was performed by AssayMAP Bravo Automated liquid handling platform (Agilent, St. Clara, CA, USA), using a pre-determined fractionation protocol. Dried peptides were reconstituted in 1% of ammonium hydroxide, and subjected to a strong anion exchange using QMA stationary phase. The stationary phase was transferred to a 5 µL resin-free cartridges. Eluted fractions were dried in a vacuum concentrator, then resuspended in 1% formic acid, and additionally fractioned by SCX stationary phase, according to the manufacturer’s protocol. Buffers used for QMA and SCX fractionation are shown in Appendix A. Finally, the resulting 36 fractions from each sample were evaporated in a vacuum concentrator.

### 4.7. Mass Spectrometry

Dried peptide fractions were resuspended in 10 µL of 5 mg/mL CHCA dissolved in 50% acetonitrile, and a volume of 1 μL was spotted onto a MALDI plate. Mass spectrometry acquisition was performed with a MALDI-TOF/TOF MS 4800 Plus analyser (AB Sciex, Framingham, MA, USA) equipped with a 200 Hz, 355 nm neodymium-doped yttrium aluminium garnet Nd: YAGlaser. Ions in MS analysis were analyzed in reflectron negative ion mode. The instrument parameters were set using the 4000 Series Explorer software V 3.5.3 (AB Sciex, Framingham, MA, USA). Mass spectra were obtained by averaging 1800 laser shots covering a mass range of *m/z* 1000 to 4000. The precursor ions generated by negative ion MS were analyzed both in negative and positive MS/MS ion mode. Tandem mass spectrometry acquisition was achieved by 1 kV collision energy, without usage of collision gas. Precursor peak selection (LC–MALDI peak MS processing) was controlled by job-wide peak selection method. Peak selection was performed automatically, in order to analyze the MS spectra and acquire MS/MS data from selected peaks. Parameters of MS/MS peak selection were as follows: minimum S/N filter = 15, minimum chromatogram peak width = 3, maximum precursors per fraction = 20, fraction-to-fraction precursor mass tolerance = 200 ppm.

### 4.8. BLASTp Parameters

Parameters of the blast search were as follows: amino acid mass tolerance = 0.3; number of queries = 10; matrix = PAM30; word size = 2; window size = 40; threshold = 11; gap cost existence = 9; gap costs extension = 1. The E-value was set to 0.1, in order to show only relevant results. Peptides were matched to a custom urinary database by an in-house-developed application written in C#, which utilizes Apache-Lucene-based search engine as it is described elsewhere [32]. The results were grouped by the number of queries (peptides) matched to a species in the database where the highest number of peptide matches correspond to an identified species.

## 5. Conclusions

We successfully applied a bottom-up mass spectrometry approach to identify bacterial species causing urinary tract infections directly from urine samples. Overall, the identification of pathogens causing urinary tract infections is achieved within 24 h, without a prior cultivation step. A novel de novo peptide sequencing software was developed to provide accurate identification even when processing a large search database. Furthermore, a mathematic model was applied to estimate the reliability of the identification result. Taxonomic resolution was satisfactory, however, it can be improved with the use of more sensitive instrumentation and curated database. In addition to urinary tract infections, the strategy used in this study could also be used for the analysis of other clinical samples such as blood cultures and stool samples, therefore, providing a path for further improvement of widely used MALDI MS techniques (MALDI MS to MALDI MS/MS) in clinical microbiology.

## Figures and Tables

**Figure 1 molecules-27-05461-f001:**
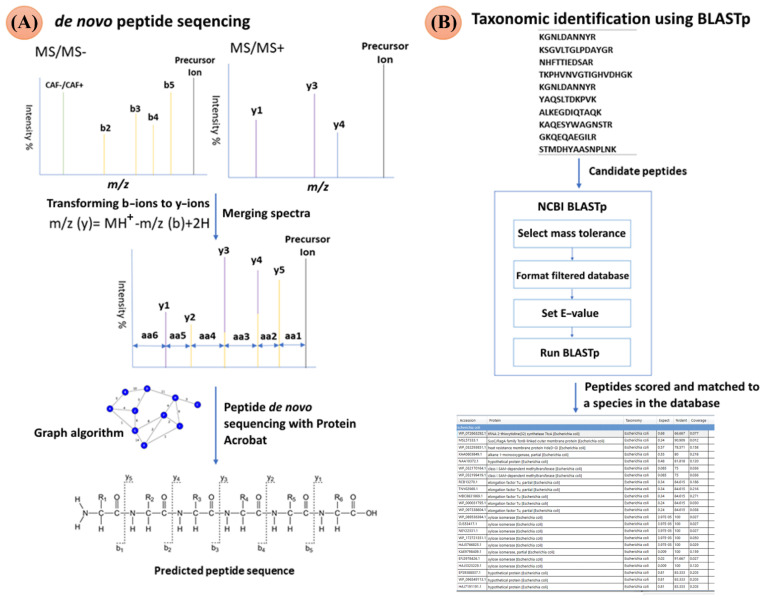
Pipeline for the identification of species by de novo peptide sequencing where (**A**) positive and negative MS/MS spectra were aligned, transformed in silico to y-ions, and used for the prediction of peptide sequence. (**B**) Elucidated peptide sequences were scored according to the highest intensities, and searched against NCBI*nr* database by BLASTp.

**Figure 2 molecules-27-05461-f002:**
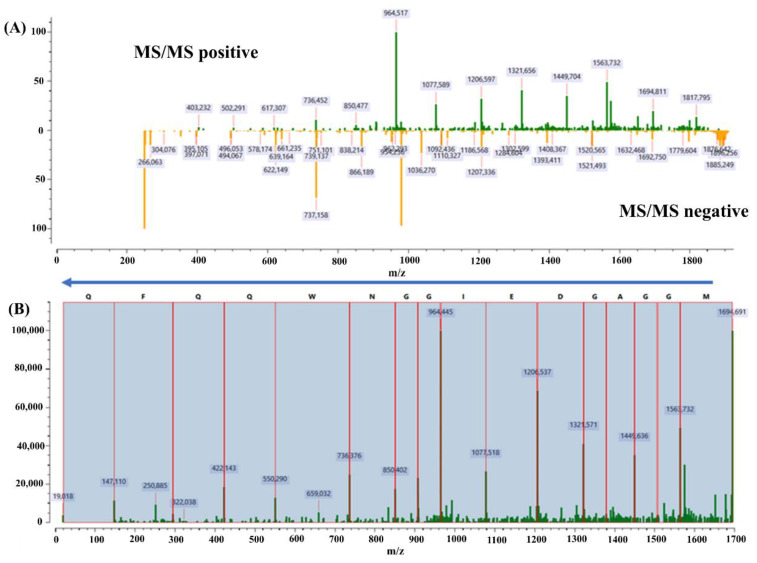
Obtained results used to identify the Putative stress protein found in *K. pneumoniae* (accession number STS794191). (**A**) The mirror image of positive and negative MS/MS spectra, where the positive spectrum (green) represents y-ions prevalently, and the negative spectrum (orange) represents mostly b-ions of the analyzed peptide. (**B**) The denoted peptide sequence was determined as “MNKDEIGGNWKQFK” by Protein Acrobat, where red lines represent matched amino acids to each ion mass.

**Figure 3 molecules-27-05461-f003:**
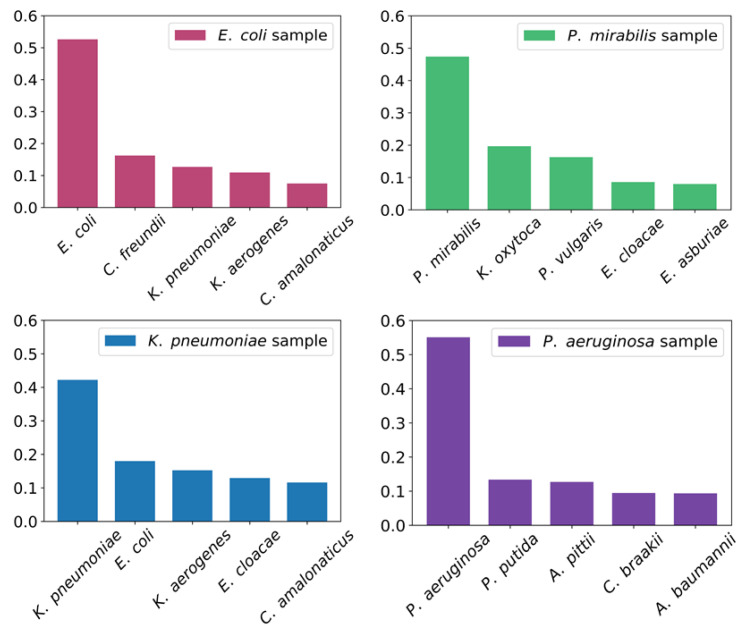
Identification from pure bacterial colonies by MALDI-TOF/TOF and de novo peptide sequencing. Results are plotted for *E. coli*, *P. mirabilis*, *K. pneumoniae*, and *P. aeruginosa*. The *Y*-axis indicates the probability of correct species identification, and the bars represent the five bacterial species with the highest number of matched peptides for each bacterium separately.

**Table 1 molecules-27-05461-t001:** The results of direct identification of uropathogenic species by tandem mass spectrometry analysis and de novo sequencing with Protein Acrobat. The results are compared to the standard urine culture test, and bolded if they are misidentified.

Standard Urine Culture	Proteomic Method	Conf. ^1^	Standard Urine Culture	Proteomic Method	Conf. ^1^
*Klebsiella pneumoniae*	*Escherichia coli*	1	*Escherichia coli*	*Escherichia coli*	3
*Klebsiella aerogenes*	/^2^	1	*Escherichia coli*	*Escherichia coli*	3
*Klebsiella pneumoniae*	*Escherichia coli*	1	*Escherichia coli*	*Escherichia coli*	3
*Escherichia coli*	*Escherichia coli*	1	*Escherichia coli*	*Escherichia coli*	3
*Candida albicans*	*Candida albicans*	2	*Escherichia coli*	*Escherichia coli*	3
*Escherichia coli*	*Escherichia coli*	2	*Escherichia coli*	*Escherichia coli*	3
*Escherichia coli*	*Escherichia coli*	2	*Escherichia coli*	*Escherichia coli*	3
*Klebsiella pneumoniae*	*Klebsiella pneumoniae*	2	*Escherichia coli*	*Escherichia coli*	3
*Enterococcus faecalis*	*Enterococcus faecalis*	2	*Enterococcus faecalis*	*Enterococcus faecalis*	3
*Proteus mirabilis*	*Proteus mirabilis*	2	*Enterococcus faecalis*	*Enterococcus faecalis*	3
*Stenotrophomonas maltophilia*	*Stenotrophomonas maltophilia*	2	*Pseudomonas aeruginosa*	*Pseudomonas aeruginosa*	3
*Citrobacter freundii*	*Klebsiella pneumoniae*	3	*Pseudomonas aeruginosa*	*Pseudomonas aeruginosa*	3
*Escherichia coli*	*Escherichia coli*	3	*Pseudomonas aeruginosa*	*Pseudomonas aeruginosa*	3
*Escherichia coli*	*Escherichia coli*	3	*Pseudomonas aeruginosa*	*Pseudomonas aeruginosa*	3
*Escherichia coli*	*Escherichia coli*	3	*Proteus mirabilis*	*Proteus mirabilis*	3
*Escherichia coli*	*Escherichia coli*	3	*Proteus mirabilis*	*Proteus mirabilis*	3
*Escherichia coli*	*Escherichia coli*	3			

^1^ Confidence level: high-confidence level score, 3; medium-confidence level score, 2; and low-confidence level score, 1. ^2^ Not identified.

## Data Availability

All data are provided in the Appendix A or are available on request.

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
