# Peer review of "Direct Identification of Urinary Tract Pathogens by MALDI-TOF/TOF Analysis and De Novo Peptide Sequencing"

_molecules, 2022, doi:10.3390/molecules27175461_

Round 1
Reviewer 1 Report
The presented work is devoted to the development of a methodology for solving the important problem of rapid urinary tract pathogens identification. Rapid identification of the infection nature may be critical in some clinical cases.
In addition to the speed of identification, the test sensitivity is also important. As shown by the authors, the approach they developed has sufficient resolution for use in clinical practice. The conclusions of the work look quite reasonable, and the results are reproducible.
Nevertheless, when discussing some negative results of the work (incorrect identification of pathogens using the developed methodology), the authors made several insufficiently substantiated assumptions.
11) Pointing in line 260 to “high genomic similarity between Klebsiella, Escherichia, and Citrobacter species”, the authors used references where this is not shown. The references should be replaced with others that will better substantiate the thesis expressed.
22) Line 258 indicates a possible reason for identification failure - coinfection with two or more pathogens. This situation can be observed in practice and, according to various estimates, is up to 20% or more of clinical cases. The authors had the opportunity to establish the presence of co-infection, since they simultaneously carried out identification by cultural methods. This data should be provided if it is not lost.
Also, at the end of line 261, there was an “or” which is not needed there.
Author Response
Point 1: Pointing in line 260 to “high genomic similarity between Klebsiella, Escherichia, and Citrobacter species”, the authors used references where this is not shown. The references should be replaced with others that will better substantiate the thesis expressed.
Response 1: The authors acknowledged the mistake of choosing reference number 35. (Brisse, S.; Verhoef, J. Phylogenetic Diversity of Klebsiella Pneumoniae and Klebsiella Oxytoca Clinical Isolates Revealed by Randomly Amplified Polymorphic DNA, GyrA and ParC Genes Sequencing and Automated Ribotyping. Int J Syst Evol Microbiol 2001, 51, 915–924, doi:10.1099/00207713-51-3-915) and is now deleted.
Although, reference number 36. (Kumar, V.; Sun, P.; Vamathevan, J.; Li, Y.; Ingraham, K.; Palmer, L.; Huang, J.; Brown, J.R. Comparative Genomics of Klebsiella Pneumoniae Strains with Different Antibiotic Resistance Profiles. Antimicrobial Agents and Chemotherapy 2011, 55, 4267–4276, doi:10.1128/AAC.00052-11) is stated:
“K. pneumoniae, a gammaproteobacterium of the family Enterobacteriaceae, is a close relative of many familiar genera, such as Citrobacter, Escherichia, Enterobacter, and Salmonella. A distinguishing characteristic of K. pneumoniae is a thick polysaccharide coat which might facilitate its evasion of host defenses. The close genetic association of the members of the family Enterobacteriaceae facilitates the inter- and intraspecies transmission of plasmids and insertion elements, which are often vectors for the horizontal exchange of antibiotic resistance genes.” and was kept in the article.
Point 2: Line 258 indicates a possible reason for identification failure - coinfection with two or more pathogens. This situation can be observed in practice and, according to various estimates, is up to 20% or more of clinical cases. The authors had the opportunity to establish the presence of co-infection, since they simultaneously carried out identification by cultural methods. This data should be provided if it is not lost. Also, at the end of line 261, there was an “or” which is not needed there.
Response 2: The authors thank the reviewer and acknowledge the observation. The authors only used pure colonies of known bacteria to validate provided identification results but used urine samples were not cultured. For the same reason, we can not provide asked results.
Reviewer 2 Report
In the work entitled “Direct identification of urinary tract pathogens by MALDI-TOF/TOF analysis and de novo peptide sequencing”, the authors used MALDI-TOF/TOF instrument and in negative and positive ion modes and de novo peptide sequencing for pathogen identification. The manuscript is well organized with necessary data and within the scope of this Journal. It has novelty and can be accept for publication.
This paper is well written and well organized. The introduction and background are reasonable given the promise of the paper. Figures and tables are comprehensive and helpful.
The paper generates the following kinds of data:
1. Pathogen identification using MALDI-TOF/TOF instrument
2. Providing chemically activated fragments for peptides analyzing
3. De novo peptide sequencing and aligning with the NCBInr
4. Developing a custom program package for data analysis that predicted 27 peptide sequences from the negative and positive MS/MS spectra
The topic is original and the main question addressed by the research over a conventional MALDI-TOF peptide analysis is developing a MAL-2 DI-TOF/TOF analysis which allows to identify pathogens directly from urinary tract and grown on agar plates in less time without a cultivation step. It is interesting and relevant on peptide identification using NCBI Database derived from genome sequencing. The conclusions are consistent with the evidence and arguments presented in the paper and the results address the main question posed.
Author Response
Authors thanks reviewer 2 for the comments and revision of the manuscript.
Reviewer 3 Report
Title: Direct identification of urinary tract pathogens by MALDI-TOF/TOF analysis and de novo peptide sequencing
Manuscript ID: molecules-1869626
The presented study presents an interesting methodology for the possibility of direct identification of pathogens caused of UTI. I consider the study to be beneficial and to provide new information on the possibilities of correct and, in principle, quickly feasible identification of the causes of UTI pathogens. Methodologically, the manuscript is well described. The results are interesting and quite well and clearly interpreted.
However, I have comments or suggestions, see below:
1/ Please unify and correctly state the entire "name", or designation of mass spectrometry methodology as it is still primarily "mass spectrometry" (MS) without the special modification used - i.e. L. 59 (MALDI-TOF MS), L. 91-92 (MALDI-TOF MS), etc. The comment refers to many places within the text, incl. results, discussion, descriptions of objects within the manuscript, etc.
2/ L. 43 – it is possible to omit "RNA" and leave directly rRNA in the sentence without brackets
3/ L. 60 please supplement the citation
4/ L. 92 "-MS/MS" please give a better expression
5/ L. 182-183, L. 192, etc. – I recommend a different expression in sentences, e.g. "...for pure colony samples of E. coli (A), P. mirabilis (B),...", etc.
6/ L. 197 – please rewrite the sentence in a more appropriate way
7/ Table 1 – explanation of the symbol "/" used is missing
8/ Table 1 – error in the given name Stenotrophomonas – error in Latin notation + occurrence of incorrect characters, etc.
9/ Discussion – the entire chapter contains many specific results of the study and, conversely, little cited literature, few comparisons with previous research. The discussion chapter must be rewritten in a suitable form.
10/ I recommend rewriting the Conclusion in a more concise form, it would also be appropriate to better state the conclusion and supplement the main findings.
I recommend language corrections by native speakers.
Author Response
Point 1: Please unify and correctly state the entire "name", or designation of mass spectrometry methodology as it is still primarily "mass spectrometry" (MS) without the special modification used - i.e. L. 59 (MALDI-TOF MS), L. 91-92 (MALDI-TOF MS), etc. The comment refers to many places within the text, incl. results, discussion, descriptions of objects within the manuscript, etc.
Response 1: The authors thank the reviewer for the remark and “MS” is added throughout the entire text.
Point 2: L. 43 – it is possible to omit "RNA" and leave directly rRNA in the sentence without brackets
Response 2: The RNA is now stated as rRNA and brackets are deleted.
Point 3: L. 60 please supplement the citation
Response 3: The authors thank the reviewer for the comment. Reference [9] is added in line 62:
Rychert, J. Benefits and Limitations of MALDI-TOF Mass Spectrometry for the Identification of Microorganisms. Journal of Infectiology and Epidemiology 2019, 2.
Point 4: L. 92 "-MS/MS" please give a better expression
Response 4: The authors stated the full name of the instrument (MALDI-TOF-MS/MS) in line 94.
Point 5: L. 182-183, L. 192, etc. – I recommend a different expression in sentences, e.g. "...for pure colony samples of E. coli (A), P. mirabilis (B),...", etc.
Response 5: The authors acknowledged the mistake as in Figure 3 results for each bacteria were not labeled as A), B),..etc. Labels were now deleted from the text.
Point 6: L. 197 – please rewrite the sentence in a more appropriate way
Response 6: The sentence in line 197 (now 199) is now written as: “To determine the accuracy of UTIs identification results, using decision tree learning defined the confidence level of bacterial identification”.
Point 7: Table 1 – explanation of the symbol "/" used is missing
Response 7: The explanation of the symbol “/” is now added in line 227.
Point 8: Table 1 – error in the given name Stenotrophomonas – error in Latin notation + occurrence of incorrect characters, etc.
Response 8: The reviewer is correct. Table 1 is now edited as suggested.
Point 9: Discussion – the entire chapter contains many specific results of the study and, conversely, little cited literature, few comparisons with previous research. The discussion chapter must be rewritten in a suitable form.
Response 9: The chapter is now supported with more literature cmparsions to put the study in a more informative context. The following changes were done:
Lines 229-233 the text and references were added: “Pathogen identification using tandem mass spectrometry is a developing methodology with a promising result. Accurate identification of samples derived from pure colonies was shown in several publications [35,36] (Alves, G.; Wang, G.; Ogurtsov, A.Y.; Drake, S.K.; Gucek, M.; Sacks, D.B.; Yu, Y.-K. Rapid Classification and Identification of Multiple Microorganisms with Accurate Statistical Significance via High-Resolution Tandem Mass Spectrometry. J. Am. Soc. Mass Spectrom. 2018, 29, 1721–1737, doi:10.1007/s13361-018-1986-y and Karlsson, R.; Davidson, M.; Svensson-Stadler, L.; Karlsson, A.; Olesen, K.; Carlsohn, E.; Moore, E.R.B. Strain-Level Typing and Identification of Bacteria Using Mass Spectrometry-Based Proteomics. J. Proteome Res. 2012, 11, 2710–2720, doi:10.1021/pr2010633). However, the main advantage of this methodology is the possibility to use directly from clinical samples, thus avoiding the lengthy culture step [37] (Karlsson, R.; Gonzales-Siles, L.; Gomila, M.; Busquets, A.; Salvà-Serra, F.; Jaén-Luchoro, D.; Jakobsson, H.E.; Karlsson, A.; Boulund, F.; Kristiansson, E.; et al. Proteotyping Bacteria: Characterization, Differentiation and Identification of Pneumococcus and Other Species within the Mitis Group of the Genus Streptococcus by Tandem Mass Spectrometry Proteomics. PLOS ONE 2018, 13, e0208804, doi:10.1371/journal.pone.0208804).”
In line 245 a reference was added to support the claim: The resulting complex peptide mixtures were separated into 36 fractions prior to mass spectra analysis to increase proteome coverage and facilitate identification of low abundance proteins (Bruno Manadas, Vera M Mendes, Jane English & Michael J Dunn (2010) Peptide fractionation in proteomics approaches, Expert Review of Proteomics, 7:5, 655-663, DOI: 10.1586/epr.10.46)
In line 271 was added supporting statement “Table S1”.
Also, reference [43] is added in line 277 (Tran, N.H.; Zhang, X.; Xin, L.; Shan, B.; Li, M. De Novo Peptide Sequencing by Deep Learning. Proc Natl Acad Sci U S A 2017, 114, 8247–8252, doi:10.1073/pnas.1705691114.) and the comparison with other similar studies can be found in the Introduction (lines 72-89).
Point 10: I recommend rewriting the Conclusion in a more concise form, it would also be appropriate to better state the conclusion and supplement the main findings. I recommend language corrections by native speakers.
Response 10: The authors thank the reviewer and the conclusion was altered to make it more conscise as “We successfully applied bottom up mass spectrometry approach to identify bacterial species causing urinary tract infections directly from urine samples. Overall, the identification of pathogens causing urinary tract infections was achieved within 24 hours without a prior cultivation step. A novel de novo peptide sequencing software was developed to provide accurate identification even when when processing large search detabase. Furthermore, a mathematic model was applied to estimate the reliability of the identification result. Taxonomic resolution was satisfactory, however it can be improved with the use of more sensitive instrumentation and curated database. In addition to urinary tract infections, the strategy used in this study could also be used for the analysis of other clinical samples such as blood cultures and stool samples, therefore providing a path for further improvement of widely used MALDI techniques (MALDI-MS and MALDI-MS/MS) in clinical microbiology”.
Round 2
Reviewer 3 Report
I am satisfied with the changes and corrections.
I recommend the paper for publication in Molecules/MDPI.